# Compress Guidance in Conditional Diffusion Sampling

## Abstract

We found that enforcing guidance throughout the sampling process is often counterproductive due to the model-fitting issue, where samples are 'tuned' to match the classifier's parameters rather than generalizing the expected condition. This work identifies and quantifies the problem, demonstrating that reducing or excluding guidance at numerous timesteps can mitigate this issue. By distributing a small amount of guidance over a large number of sampling timesteps, we observe a significant improvement in image quality and diversity while also reducing the required guidance timesteps by nearly 40%. This approach addresses a major challenge in applying guidance effectively to generative tasks. Consequently, our proposed method, termed Compress Guidance, allows for the exclusion of a substantial number of guidance timesteps while still surpassing baseline models in image quality. We validate our approach through benchmarks on label-conditional and text-to-image generative tasks across various datasets and models.

## 1 Introduction

Guidance in diffusion models is mainly divided into classifier-free guidance in Ho & Salimans (2022), and classifier guidance in Dhariwal & Nichol (2021). Although both of these methods significantly improve the performance of the diffusion samples Dhariwal & Nichol (2021); Ho & Salimans (2022); Bansal et al. (2023); Liu et al. (2023); Epstein et al. (2023), they both suffer from large computation time. For classifier guidance, the act of gradients calculation backwards through a classifier is costly. On the other hand, forwarding through a diffusion model twice at every timestep also costs significant computation in classifier-free guidance.

This work challenges the necessity of the current complex process based on several key observations. First, we find that the guidance loss is predominantly active during the early stages of the sampling process, when the image lacks a well-defined structure. As the model progresses and shifts its focus to refining image details, the guidance loss tends to approach zero. Additionally, when evaluating intermediate samples with an additional classifier not used for guidance, we observe that the loss from this external classifier does not decrease in the same way as it does for the guidance-specific classifier. This suggests that the generated samples are tailored to fit the features of the guiding classifier rather than producing generalized features applicable to different classifiers. We define this issue as *model-fitting*, where the generated image pixels are optimized to satisfy the guiding classifier's criteria rather than generalizing to the intended conditions. The problem is validated by three pieces of evidence in section 3.1.

These observations prompt us to question whether guidance is necessary at every timestep and how reducing the frequency of guidance could enhance generative quality. In Section 3.2, we further explore the properties of guidance in ensuring sample quality. Based on this analysis, we propose a simple yet effective method called Compress Guidance (CompG), which mitigates the issue by reducing the number of timesteps that invoke gradient calculation. This approach not only improves sample quality but also significantly accelerates the overall process as shown in Fig.1.

Overall, the contributions of our works are three-fold: **(1)** Explore and quantify the model-fitting problem in guidance and the redundant computation resulting from current guidance methods. **(2)** Propose a simple but effective method to contain the model-fitting problem and improve computational time. **(3)** Extensive analysis and experimental results for different datasets and generative tasks on both classifier and classifier-free guidance perspectives.

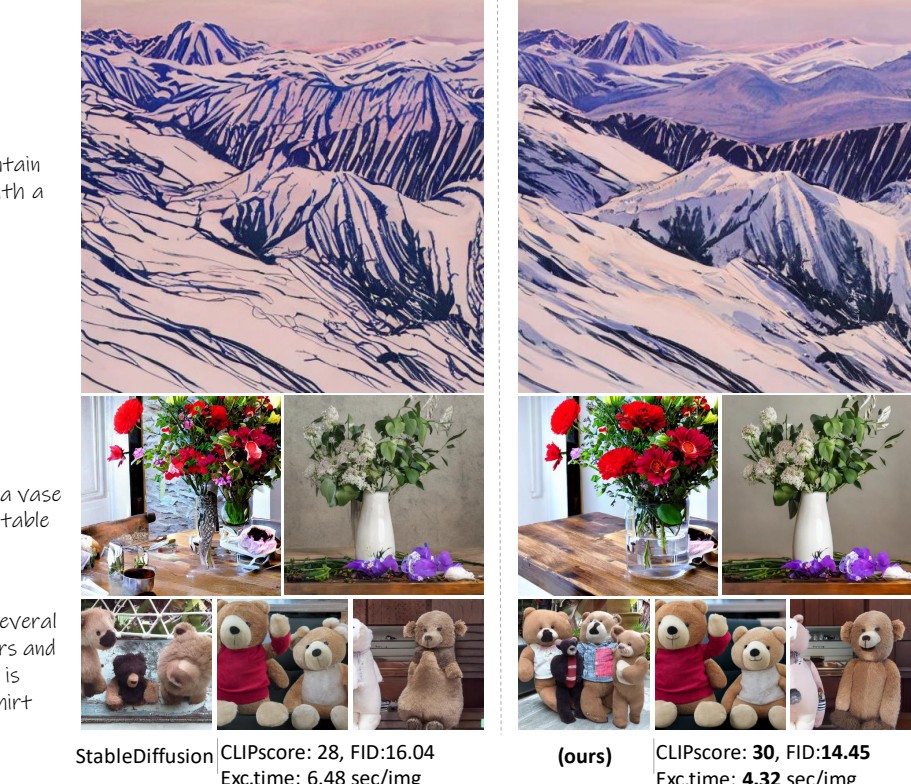

Serene mountain landscape with a clear sky

Flowers are arranged in a vase sitting on a table

There are several stuffed bears and one of them is wearing a shirt

StableDiffusion | CLIPscore: 28, FID:16.04 Exc.time: 6.48 sec/img

(ours) | CLIPscore: **30**, FID:**14.45** Exc.time: **4.32** sec/img

Figure 1: *Stable Diffusion with classifier-free guidance. The left figure is the vanilla classifier-free guidance with application on all 50 timesteps. Our proposed Compress Guidance method is the right figure, where we only apply guidance on 10 over 50 steps. The output shows our methods' superiority over classifier-free guidance regarding image quality, quantitative performance and efficiency. The efficiency is counted based on the time to generate 30000 images with 1 GPU.*

## 2 BACKGROUND

**Diffusion Models** Ho et al. (2020) has the form of: $p_\theta := p(\mathbf{x}_T) \prod_{t=1}^{T} p_\theta(\mathbf{x}_{t-1}|\mathbf{x}_t)$ where $p_\theta(\mathbf{x}_{t-1}|\mathbf{x}_t) := \mathcal{N}(\mathbf{x}_{t-1}; \mu_\theta(x_t, t), \Sigma_\theta(x_t, t))$ supporting the reverse process from $\mathbf{x}_T$ to $\mathbf{x}_0$. This process is denoising process where starting from the $\mathbf{x}_T \sim \mathcal{N}(\mathbf{x}_T; 0, \mathbf{I})$ to gradually move to $\mathbf{x}_0 \sim q(\mathbf{x}_0)$. This process is trained to be matched with the forward diffusion process $q(\mathbf{x}_{1:T}|\mathbf{x}_0) := \prod_{t=1}^{T} q(\mathbf{x}_t|\mathbf{x}_{t-1})$ given $q(\mathbf{x}_t|\mathbf{x}_{t-1})$ as $q(\mathbf{x}_t|\mathbf{x}_{t-1}) := \mathcal{N}(\mathbf{x}_t; \sqrt{1-\beta_t}\mathbf{x}_{t-1}, \beta\mathbf{I})$ or we can write the conditional distribution of $\mathbf{x}_t$ given $\mathbf{x}_0$ as below:

$$q(\mathbf{x}_t|\mathbf{x}_0) := \mathcal{N}(\mathbf{x}_t; \sqrt{\bar{\alpha}_t}\mathbf{x}_0, (1-\bar{\alpha})\mathbf{I}) \tag{1}$$

$\beta_t$ is the fixed variance scheduled before the process starts, Ho et al. (2020) denotes $\alpha_t := 1 - \beta_t$ and $\bar{\alpha}_t := \prod_{s=1}^{t} \alpha_s$ used in Eq.1. We have the $\mathbf{x}_{t-1}$ conditioned on $\mathbf{x}_0$ and $\mathbf{x}_t$ as:

$$q(\mathbf{x}_{t-1}|\mathbf{x}_t, \mathbf{x}_0) = \mathcal{N}(\mathbf{x}_{t-1}; \tilde{\boldsymbol{\mu}}_t(\mathbf{x}_t, \mathbf{x}_0), \tilde{\beta}_t\mathbf{I}) \tag{2}$$

where $\tilde{\boldsymbol{\mu}}_t(\mathbf{x}_t, \mathbf{x}_0) := \frac{\sqrt{\bar{\alpha}_{t-1}}\beta_t}{1-\bar{\alpha}_t}\mathbf{x}_0 + \frac{\sqrt{\alpha_t}(1-\bar{\alpha}_{t-1})}{1-\bar{\alpha}_t}\mathbf{x}_t$ and $\tilde{B}_t := \frac{1-\bar{\alpha}_{t-1}}{1-\bar{\alpha}_t}\beta_t$. To train the diffusion model, the lower bound loss is utilized as below:

$$\mathbb{E}[-\log p_\theta(\mathbf{x}_0)] \leq \mathbb{E}_q[-\log p(\mathbf{x}_T) - \Sigma_{t\geq 1}\log\frac{p_\theta(\mathbf{x}_{t-1}|\mathbf{x}_t)}{q(\mathbf{x}_t|\mathbf{x}_{t-1})}] \tag{3}$$

Rewrite Eq. 3 as $\mathbb{E}_q[D_{KL}(q(\mathbf{x}_T|\mathbf{x}_0)||p(\mathbf{x}_T)) + \sum_{t>1} D_{KL}(q(\mathbf{x}_{t-1}|\mathbf{x}_t, \mathbf{x}_0)||p_\theta(\mathbf{x}_{t-1}|\mathbf{x}_t)) - \log p_\theta(\mathbf{x}_0|\mathbf{x}_1)]$ The training process actually optimize the $\sum_{t>1} D_{KL}(q(\mathbf{x}_{t-1}|\mathbf{x}_t, \mathbf{x}_0)||p_\theta(\mathbf{x}_{t-1}|\mathbf{x}_t))$ where the diffusion model try to match the distribution of $\mathbf{x}_{t-1}$ by using only $\mathbf{x}_t$. There are several implementations for optimising the 3. However, the $\theta$ as parameters of the noise predictor $\epsilon_\theta(\mathbf{x}_t, t)$ is the most popular choice. After the $\theta$ are trained using Eq. 3, the sampling equation:

$$\mathbf{x}_{t-1} = \frac{1}{\sqrt{\alpha_t}}(\mathbf{x}_t - \frac{1-\alpha_t}{\sqrt{1-\bar{\alpha}_t}}\epsilon_\theta(\mathbf{x}_t, t)) + \sigma_t\mathbf{z} \tag{4}$$

**Guidance** in the Diffusion model offers conditional information and image quality enhancement. Given a classifier $p_\phi(y|\mathbf{x}_t)$ that match with the labels distribution conditioned on images $\mathbf{x}_t$, we have the sampling equation with guidance as:

$$\mathbf{x}_{t-1} \sim \mathcal{N}(\mu_t + s\sigma_t^2 \nabla_{\mathbf{x}_t} \log p_\phi(y|\mathbf{x}_t), \sigma_t) \tag{5}$$

with $s$ is the guidance scale. Beside the classifier guidance as Eq.5, Ho & Salimans (2022) proposes another version named classifier-free guidance. This guidance method does not base the information on a classifier. Instead, the guidance depends on the conditional information from a conditional diffusion model. The sampling equation has the form:

$$\mathbf{x}_{t-1} \sim \mathcal{N}(\tilde{\boldsymbol{\mu}}_t(\mathbf{x}_t, \frac{\mathbf{x}_t - \sqrt{1-\bar{\alpha}}\tilde{\epsilon}_t}{\sqrt{\bar{\alpha}_t}}), \sigma_t) \tag{6}$$

given $\tilde{\epsilon} = (1+w)\epsilon_\theta(\mathbf{x}_t, c) - w\epsilon_\theta(\mathbf{x}_t)$ with $w$ is the guidance scale.

## 3 Model-fitting in Guidance

We begin by modelling the sampling equation as two distinct optimization objectives, illustrating that the sampling process functions as a form of "training", where parameters $\mathbf{x}_t$ are optimized over $T$ timesteps. We then analyze the "training" of $\mathbf{x}_t$ in light of these objectives, highlighting the model-fitting problem that arises in the current guidance-driven sampling process. To address this issue, we propose a simple method called Compress Guidance, which helps mitigate the observed model-fitting problem. From Eq.4, we have:

$$\mathbf{x}_{t-1} = \frac{(1-\alpha_t)\sqrt{\bar{\alpha}_{t-1}}}{1-\bar{\alpha}_t} \frac{\mathbf{x}_t - \sqrt{1-\bar{\alpha}_t}\epsilon_\theta(\mathbf{x}_t, t)}{\sqrt{\bar{\alpha}_t}} + \frac{(1-\bar{\alpha}_{t-1})\sqrt{\alpha_t}}{1-\bar{\alpha}_t}\mathbf{x}_t + \sigma_t z \tag{7}$$

**Distribution matching objective**: Assuming that $\epsilon_\theta(\mathbf{x}_t, t)$ is learned perfectly to match random noise $\epsilon$ at timestep $t$, we have $\frac{\mathbf{x}_t - \sqrt{1-\bar{\alpha}_t}\epsilon_\theta(\mathbf{x}_t,t)}{\sqrt{\bar{\alpha}_t}} = \mathbf{x}_0$ is the exact prediction of $\mathbf{x}_0$ at timestep $t$ according to Eq.1. With $\tilde{\mathbf{x}}_0$ is the prediction of $\mathbf{x}_0$ at timestep $t$, we can re-write the equation as bellow:

$$\mathbf{x}_{t-1} = \frac{(1-\alpha_t)\sqrt{\bar{\alpha}_{t-1}}}{1-\bar{\alpha}_t}\tilde{\mathbf{x}}_0 + \frac{(1-\bar{\alpha}_{t-1})\sqrt{\alpha_t}}{1-\bar{\alpha}_t}\mathbf{x}_t + \sigma_t z \tag{8}$$

This equation 8 can be derived from $q(\mathbf{x}_{t-1}|\mathbf{x}_t, \mathbf{x}_0)$ in Eq. 2 with parameterized trick for Gaussian Distribution. Thus, the first aim of the sampling process is to match the distribution $q(\mathbf{x}_{t-1}|\mathbf{x}_t, \mathbf{x}_0)$. Nevertheless, the Eq.8 is based on the assumption that $\tilde{\mathbf{x}}_0 \sim \mathbf{x}_0$, which often does not hold when $t \to T$. Given $\tilde{\mathbf{x}}_0 = \frac{\mathbf{x}_t - \sqrt{1-\bar{\alpha}_t}\epsilon_\theta(\mathbf{x}_t,t)}{\sqrt{\bar{\alpha}_t}}$, this formulation is rooted from $\tilde{\mathbf{x}}_0 \sim \mathcal{N}(\frac{1}{\sqrt{\bar{\alpha}}}\mathbf{x}_t; \frac{\bar{\alpha}-1}{\bar{\alpha}}\mathbf{I})$ with assumption that $\epsilon_\theta(\mathbf{x}_t, t) \sim \epsilon$. However, $\epsilon_\theta(\mathbf{x}_t, t)$ is trained to minimize $D_{KL}[q(\mathbf{x}_{t-1}|\mathbf{x}_t, \mathbf{x}_0)||p_\theta(\mathbf{x}_{t-1}|\mathbf{x}_t)]$ as in Ho et al. (2020) which actually causes a significantly distorted information if $\epsilon_\theta(\mathbf{x}_t, t)$ is utilized to sample $\tilde{\mathbf{x}}_0$ from $\mathbf{x}_t$ if $t \to T$. A smaller $t$ would result in a better prediction of $\mathbf{x}_0$ and with $t = 0$, we have $\bar{\alpha} = 1$ resulting in $\tilde{\mathbf{x}}_0 = \mathbf{x}_t$.

**Theorem 1.** *Assume that $\epsilon_\theta$ is trained to converge and the real data density function $q(\mathbf{x}_0)$ satisfies a form of Gaussian distribution. The process of recurrent sampling $\mathbf{x}_{t-1} \sim q(\mathbf{x}_{t-1}|\mathbf{x}_t, \tilde{\mathbf{x}}_0)$ from $T$ to 0 is equivalent to minimization process of $D_{KL}[q(\mathbf{x}_0)||p_\theta(\tilde{\mathbf{x}}_0|\mathbf{x}_t)]$ .wrt. $\mathbf{x}_t$.*

*Proof.* Given real data $\mathbf{x}_0$, two latent samples are sampled at two timestep $t_1 < t_2$. We have, $\mathbf{x}_{t_1} = \sqrt{\bar{\alpha}_{t_1}}\mathbf{x}_0 + \sqrt{1-\bar{\alpha}_{t_1}}\epsilon$ and $\mathbf{x}_{t_2} = \sqrt{\bar{\alpha}_{t_2}}\mathbf{x}_0 + \sqrt{1-\bar{\alpha}_{t_2}}\epsilon$. From $\mathbf{x}_{t_1}$ and $\mathbf{x}_{t_2}$, real data prediction has the form of $\tilde{\mathbf{x}}_0^{(t_1)} = \frac{\mathbf{x}_{t_1} - \sqrt{1-\bar{\alpha}_{t_1}}\epsilon_\theta(\mathbf{x}_{t_1}, t_1)}{\sqrt{\bar{\alpha}_{t_1}}}$ and $\tilde{\mathbf{x}}_0^{(t_2)} = \frac{\mathbf{x}_{t_2} - \sqrt{1-\bar{\alpha}_{t_2}}\epsilon_\theta(\mathbf{x}_{t_2}, t_2)}{\sqrt{\bar{\alpha}_{t_2}}}$ correspondingly. Replace $\mathbf{x}_{t_1}$ and $\mathbf{x}_{t_2}$ with $\mathbf{x}_0$ and $\epsilon$, we have $\tilde{\mathbf{x}}_0^{(t_1)} = \mathbf{x}_0 + \frac{\sqrt{1-\bar{\alpha}_{t_1}}(\epsilon - \epsilon_\theta(\mathbf{x}_{t_1}, t_1))}{\sqrt{\bar{\alpha}_{t_1}}}$ and $\tilde{\mathbf{x}}_0^{(t_2)} = \mathbf{x}_0 + \frac{\sqrt{1-\bar{\alpha}_{t_2}}(\epsilon - \epsilon_\theta(\mathbf{x}_{t_2}, t_2))}{\sqrt{\bar{\alpha}_{t_2}}}$. Thus $||\tilde{\mathbf{x}}_0^{(t_1)} - \mathbf{x}_0|| = \frac{1-\bar{\alpha}_{t_1}||\epsilon - \epsilon_\theta(\mathbf{x}_{t_1}, t_1)||}{\bar{\alpha}_{t_1}}$ and $||\tilde{\mathbf{x}}_0^{(t_2)} - \mathbf{x}_0|| = \frac{1-\bar{\alpha}_{t_2}||\epsilon - \epsilon_\theta(\mathbf{x}_{t_2}, t_2)||}{\bar{\alpha}_{t_2}}$. Since $\epsilon_\theta(\mathbf{x}_{t_1}, t_1) \sim \epsilon_\theta(\mathbf{x}_{t_2}, t_2) \sim \epsilon$, $||\epsilon - \epsilon_\theta(\mathbf{x}_{t_1}, t_1)|| \approx ||\epsilon - \epsilon_\theta(\mathbf{x}_{t_2}, t_2)|| \approx \Delta$. This results in $||\tilde{\mathbf{x}}_0^{(t_1)} - \mathbf{x}_0|| = \frac{1-\bar{\alpha}_{t_1}}{\bar{\alpha}_{t_1}}\Delta$ and $||\tilde{\mathbf{x}}_0^{(t_2)} - \mathbf{x}_0|| = \frac{1-\bar{\alpha}_{t_2}}{\bar{\alpha}_{t_2}}\Delta$.

| Evaluation Model | Accuracy |
|---|---|
| *On-sampling classifier* | 90.8% |
| *Off-sampling classifier* | 62.5% |
| *Off-sampling Resnet152* | 34.2% |

Table 1: *A significant gap exists between the on-sampling and the off-sampling classifier in terms of accuracy.*

Figure 2: *(left) OADM-C, (right) Resnet152 off-sampling loss. The On-sampling loss converges very early while leaving the off-sampling loss converges at the end of the process after the conclusion of the denoising process.*

$||\tilde{\mathbf{x}}_0^{(t_1)} - \mathbf{x}_0|| < ||\tilde{\mathbf{x}}_0^{(t_1)} - \mathbf{x}_0||$ since $\frac{1-\bar{\alpha}_{t_2}}{\bar{\alpha}_{t_2}} > \frac{1-\bar{\alpha}_{t_1}}{\bar{\alpha}_{t_1}} \geq 0, \forall t_2 > t_1$. Consequently, the sampling of $\mathbf{x}_{t-1} \sim q(\mathbf{x}_{t-1}|\mathbf{x}_t, \tilde{\mathbf{x}}_0)$ from timesteps $T$ to $0$ would mean the minimization of $||\tilde{\mathbf{x}}_0^{(t)} - \mathbf{x}_0||$. Since $q(\mathbf{x}_0)$ is a normal distribution, the final objective can be written as $\min_{\mathbf{x}_t} D_{KL}[q(\mathbf{x}_0)||p_\theta(\tilde{\mathbf{x}}_0|\mathbf{x}_t)]$. (Full proof can be found in the appendix). $\square$

If we consider $\mathbf{x}_t$ of the Eq.8 as the set of optimization parameters, the sampling process will have the objective function:

$$\min_{\mathbf{x}_t} D_{KL}[q(\mathbf{x}_0)||p_\theta(\tilde{\mathbf{x}}_0|\mathbf{x}_t)] \tag{9}$$

We re-write the Eq.8 as:

$$\mathbf{x}_{t-1} = \mathbf{x}_t - \underbrace{(\frac{\sqrt{\alpha_t}-1}{\sqrt{\alpha_t}}\mathbf{x}_t + \frac{1-\alpha_t}{\sqrt{1-\bar{\alpha}_t}\sqrt{\alpha_t}}\epsilon_\theta(\mathbf{x}_t, t) - \sigma_t \mathbf{z})}_{\gamma_1 \nabla D_{KL}[q(\mathbf{x}_0)||p_\theta(\tilde{\mathbf{x}}_0|\mathbf{x}_t)]} \tag{10}$$

Eq.10 turns the sampling process into a stochastic gradient descent process where the $\mathbf{x}_t$ is the parameter of the model at the timestep $t$, the updated direction into $\mathbf{x}_t$ aims to satisfy the objective function Eq.9.

**Classification objective**: From Eq.5, we have the term $s\sigma_t^2 \nabla_{\mathbf{x}_t} \log p_\phi(y|\mathbf{x}_t)$ is added to the sampling equation for guidance. This term can be written in full form as $s\sigma_t^2 \nabla_{\mathbf{x}_t}(q(y)\log q(y) - q(y)\log p_\phi(y|\mathbf{x}_t))$ which is equivalent to $-s\sigma_t^2 \nabla D_{KL}[q(y)||p_\phi(\hat{y}|\mathbf{x}_t)]$. Combine Eq.10 with guidance information in Eq.5, we have:

$$\mathbf{x}_{t-1} = \mathbf{x}_t - \underbrace{(\frac{\sqrt{\alpha_t}-1}{\sqrt{\alpha_t}}\mathbf{x}_t + \frac{1-\alpha_t}{\sqrt{1-\bar{\alpha}_t}\sqrt{\alpha_t}}\epsilon_\theta(\mathbf{x}_t, t) - \sigma_t \mathbf{z})}_{\gamma_1 \nabla D_{KL}[q(\mathbf{x}_0)||p_\theta(\tilde{\mathbf{x}}_0|\mathbf{x}_t)]} - \underbrace{(-s\sigma_t^2 \nabla_{\mathbf{x}_t} \log p_\phi(y|\mathbf{x}_t))}_{\gamma_2 \nabla D_{KL}[q(y)||p_\phi(\hat{y}|\mathbf{x}_t)]} \tag{11}$$

As a result, the process of updating $\mathbf{x}_t$ to $\mathbf{x}_{t-1}$ is a "training" step to optimize to objective functions $D_{KL}[q(\mathbf{x}_0)||p_\theta(\tilde{\mathbf{x}}_0|\mathbf{x}_t)]$ and $D_{KL}[q(y)||p_\phi(\hat{y}|\mathbf{x}_t)]$ with two gradients respecting to $\mathbf{x}_t$ as Eq.11. Since this is similar to the training process, it is expected to face some problems in training deep neural networks. In this work, the problem of model fitting is detected by observing the losses given by the classification objective during the sampling process.

### 3.1 MODEL-FITTING

Based on the optimization problem from the sampling process in the previous section, we first define *on-sampling loss* and *off-sampling loss* for observation.

**Definition 1.** *On-sampling loss/accuracy refers to the loss or accuracy evaluated on the generated samples $\mathbf{x}_t$ at timestep $t$ during the diffusion sampling process, which consists of $T$ timesteps. This loss is defined as $-\log p_\phi(\hat{y}|\mathbf{x}_t)$ by the classifier parameters $\phi$ that provides guidance throughout the sampling process.*

**Definition 2.** *Off-sampling loss/accuracy refers to the loss or accuracy evaluated on the generated samples $\mathbf{x}_t$ at timestep $t$ during the diffusion sampling process, which consists of $T$ timesteps. This loss is defined as $-\log p_{\phi'}(\hat{y}|\mathbf{x}_t)$ by the classifier parameters $\phi'$ that **does not** provides guidance throughout the sampling process.*

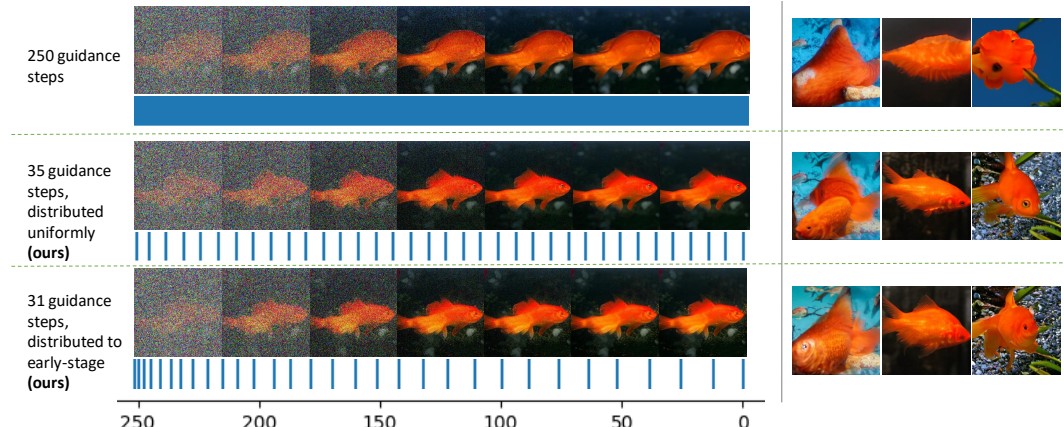

**Figure 3:** *ImageNet256x256 samled by ADM-G in Dhariwal & Nichol (2021). The top row is the vanilla guidance, where all the timesteps got the guidance information. The second and third rows are our proposed method, which only applies 35 time steps. The second row distributes the timesteps uniformly, while the third row distributes the timesteps toward the early stage of the sampling process. The Compress Guidance performs significantly better than the original guidance method. One blue stick means one guidance step.*

we visualize the *on-sampling* loss obtained from the noise-aware ADM classifier in Dhariwal & Nichol (2021) as in Figure 2. We found out that the classification information is mainly active during the early stage of the process, as it converges very early in the first 120 timesteps. However, a different trend is observed for the *off-sampling* loss. We set up an off-sampling classifier with the same architecture and performance as the on-sampling classifier used for guidance or in *on-sampling* loss. The only difference between the two models is the parameters. The details on obtaining this off-sampling classifier are in Appendix B. We name this off-sampling classifier as OADM-C. To avoid bias, we also use an off-the-shelf model ResNet152 He et al. (2015) to be another off-sampling classifier.

**Definition 3.** *Model-fitting occurs when sampled images $\mathbf{x}_t$ at timestep $t$ is updated to maximize $p_\phi(y|\mathbf{x}_t)$ or to satisfy the parameters of the $\phi$ only instead of the real distribution $q(y|\mathbf{x}_t)$.*

In practice, a pretrained $p_\phi(y|\mathbf{x}_t)$ is only able to capture part of the $q(y|\mathbf{x}_t)$. Fitting solely with $p_\phi(y|\mathbf{x}_t)$ limits the sample's generalisation ability, leading to incorrect features or overemphasising certain details due to misclassification or overfocusing of the guidance classifier. Three pieces of evidence support that the vanilla guidance suffers from **model-fitting** problem.

**Evidence 1:** From the figures in Table 2, we see that while the on-sampling loss converges around the $120^{th}$ timestep, the off-sampling loss remains high until the diffusion model converges later. This indicates that samples $\mathbf{x}_t$ at timestep $t$ satisfy only the on-sampling classifier but not the off-sampling classifier, despite their identical performance and architecture. Although the off-sampling loss decreases by the end, a significant gap between the off-sampling and on-sampling losses persists. This supports our hypothesis that the guidance sampling process produces features that fit only the guidance classifier, not the conditional information.

**Evidence 2:** Table 2 illustrates the model-fitting problem through accuracy metrics. With vanilla guidance, the accuracy is about 90.80% for the on-sampling classifier. However, the same samples evaluated by the off-sampling classifier or Resnet152 achieve only around 62.5% and 34.2% accuracy, respectively. This indicates that many features generated by the model are specific to the guidance classifier and do not generalize to other models.

**Evidence 3:** Figure 3 (first row) shows samples from vanilla guidance, where every sampling step receives guidance information. Applying guidance at all timesteps forces the model to fit the on-sampling classifier's perception. Often, this makes the model colour-sensitive, focusing on generating the "orange" feature for Goldfish and ignoring other details.

From the three pieces of evidence we can observe, we can conclude that the vanilla guidance scheme has suffered from the model-fitting problem.

**Analogy to overfitting:** In neural network training, we have a dataset $\mathbf{x}$ and a classifier $f_\theta(\mathbf{x})$ to approximate the posterior distribution $p(y|\mathbf{x})$. Let $\mathbf{x}_{\text{train}}$ be the training data and $\mathbf{x}_{\text{test}}$ the testing data.

Overfitting occurs when $f_\theta$ is tailored to fit $\mathbf{x}_{\text{train}}$ but fails to generalize to the entire dataset $\mathbf{x}$. This is observed by the gap between training loss/accuracy and testing loss/accuracy on $\mathbf{x}_{\text{train}}$ and $\mathbf{x}_{\text{test}}$.

Table 2: *Overfitting vs. Model-Fitting*

| Aspect | Overfitting | Model-fitting |
|--------|-------------|---------------|
| **Train Data** | $\mathbf{x}_{\text{train}}$ | $f_{\phi_g}$ |
| **Test Data** | $\mathbf{x}_{\text{test}}$ | $f_{\phi_o}$ |
| **Parameters** | $f_\phi$ | $\mathbf{x}$ |

In the diffusion model's sampling process, the classifier $f_\theta$ is pretrained or fixed. The aim is to adjust the samples $\mathbf{x}$ to match the trained posterior $p_\theta(y|\mathbf{x})$. This process also uses Stochastic Gradient Descent with different roles: $f_{\phi_g}$ acts as the fixed data, and $\mathbf{x}$ are the trainable parameters. The model-fitting problem arises when $\mathbf{x}$ is adjusted to fit only the specific $f_\theta$ instead of generalizing well. Here, $f_{\phi_g}$ is the on-sampling "data", and we use an off-sampling "data" $f_{\phi_o}$ to observe the model-fitting where the gap between them is large, analogous to using training and testing data to check for overfitting.

## 3.2 ANALYSIS

Gradient over-calculation is the main reason for model-fitting. Thus, **gradient balance**, which is to call not too many times of gradient calculation, is required. A straightforward solution is to eliminate the gradient calculations for the later timesteps, which have been found to be less active, as shown in Figure 2. This approach is referred to as Early Stopping (ES), where guidance is halted from the $200^{th}$ timestep onwards, continuing until the $0^{th}$ timestep.

**Early Stopping**: Figure 4 demonstrates that ES suffers from the *forgetting* problem, where on-sampling classification loss increases during the remaining sampling process, negatively impacting the generative outputs. This suggests that the guidance requires the property of **continuity**, meaning the gap between consecutive guidance steps must not be too large to prevent the *forgetting* problem.

**Uniform skipping guidance**: We try an alternative approach which is called Uniform Skipping Guidance (UG). In UG, 50 guidance steps are evenly distributed across 250 sampling steps, with guidance applied every five steps. This ensures continuity throughout the sampling process, mitigating the *forgetting* problem. However, as shown in Figure 2, UG encounters the issue of *non-convergence*, where the classification magnitude is too weak and becomes overshadowed by the denoising signals from the diffusion models, leading to poor conditional information. Thus, a guidance must require another property, which is **magnitude sufficiency**.

In summary, vanilla guidance faces the issue of *model-fitting*, while ES and UG fail due to the *forgetting* and *non-convergence* problems, respectively. Therefore, the primary goal of our proposed method is to meet three key conditions which are **gradient balance**, **guidance continuity** and **magnitude sufficiency**.

## 3.3 COMPRESS GUIDANCE

To avoid calculating too much gradient, we propose to utilize the gradient from the previous guidance step at several next sampling steps, given that the gradient magnitude difference between two consecutive sampling steps is not too significant. By doing this, we can satisfy **magnitude sufficiency** without re-calculating the gradient at every sampling step. Note that the gradient directions have not been updated since the last guidance step, resulting in the **gradient balance**. Since all the sampling

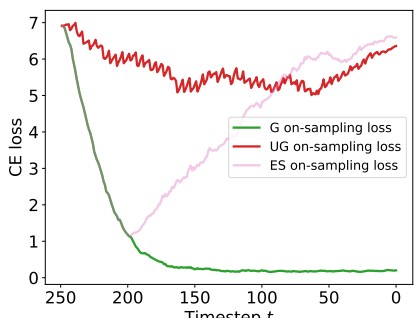

Figure 4: *G is denoted for vanilla guidance, UG is the uniform skipping scheme, and ES is the early stopping scheme. The graph shows that UG suffers from the non-convergence problem, and ES suffers from the forgetting problem.*

step receives a guidance signal, the **continuity** is guaranteed. Start with the Eq. 11, we have the sampling scheme as below:

$$\mathbf{x}_{t-1} = \begin{cases} \mathbf{x}_t - \gamma_1 \nabla D_{KL}[q(\tilde{\mathbf{x}}_0|\mathbf{x}_t)||q(\mathbf{x}_0)] - \gamma_2 \nabla D_{KL}[q(\hat{y}|\mathbf{x}_t)||q(y)], & \text{if } t \in G \\ \mathbf{x}_t - \gamma_1 \nabla D_{KL}[q(\tilde{\mathbf{x}}_0|\mathbf{x}_t)||q(\mathbf{x}_0)] - \gamma_2 \Gamma_t, & \text{otherwise} \end{cases} \quad (12)$$

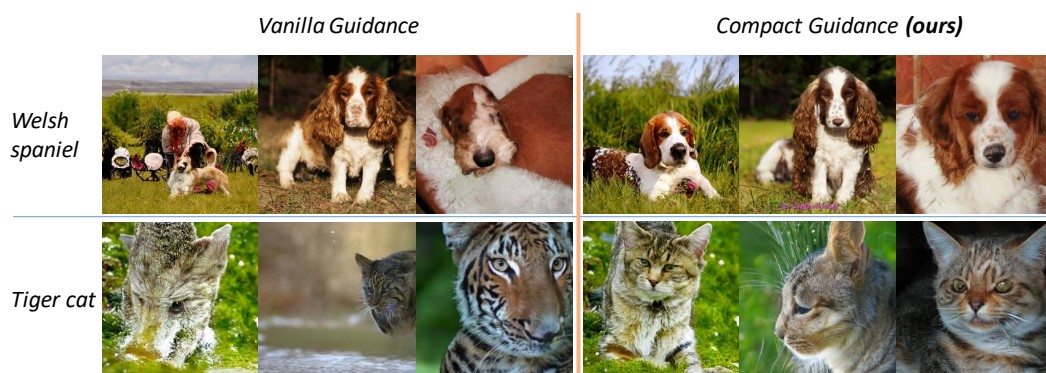

Figure 5: *Qualitative results on ImageNet256x256. Left: Vanilla guidance applied at all timesteps. Right: Compress Guidance applied at 50 out of 250 timesteps. Compress Guidance reduces over-emphasized features, correcting weird and incorrect details. Further results are in AppendixH*

The set $G$ is the set of time-steps for which the gradient will be calculated. $\Gamma$ is a variable used to store the calculated gradient from the previous sampling step, $\Gamma_t$ is updated as:

$$\Gamma_{t-1} = \begin{cases} \nabla D_{KL}[q(\hat{y}|\mathbf{x}_t)||q(y)], & \text{if } t \in G \\ \Gamma_t & \text{otherwise} \end{cases} \tag{13}$$

In practice, we find out that instead of duplicating gradients as in Eq. 12, we can slightly improve the performance by compressing the duplicated gradients into one guidance step instead of providing guidance to all sampling as in Eq.12. We name this method as *Compress Guidance.*We modify the sampling equation as below:

$$\mathbf{x}_{t-1} = \begin{cases} \mathbf{x}_t - \gamma_1 \nabla D_{KL}[q(\tilde{\mathbf{x}}_0|\mathbf{x}_t)||q(\mathbf{x}_0)] - \gamma_2 \sum_{t=G_i}^{G_{i+1}} \Gamma_t, & \text{if } t = a_i \\ \mathbf{x}_t - \gamma_1 \nabla D_{KL}[q(\tilde{\mathbf{x}}_0|\mathbf{x}_t)||q(\mathbf{x}_0)], & \text{otherwise} \end{cases} \tag{14}$$

One of the algorithm's assumptions is that the magnitude is mostly the same for two consecutive sampling steps. From Appendix G, we observe that the classification gradient magnitude difference between two consecutive sampling steps is often larger in the early stage of the sampling process. Thus, we propose a method that distributes more guidance toward the early sampling stage and sparely at the end of the process. This will help to avoid the significant accumulation of magnitude differences in the early stage and helps to deliver better performance as well as reducing the number of guidance steps. The scheme is defined as Eq. 15.

$$G_i = T - \lfloor \frac{T}{|G|^k} i^k \rfloor \quad \forall 0 \leq i \leq l, k \in [0; +\infty] \tag{15}$$

**Theorem 2.** *When $k \to +\infty$, the guidance timesteps will be distributed more toward the early stage of the sampling process.*

**Theorem 3.** *When $k < 1$ and $k \to 0$, the guidance timesteps will be distributed more toward the late stage of the sampling process.*

The proposed solution to select the timesteps for guidance as Eq.15 allows us to choose the number of timesteps we will do guidance and how to distribute these timesteps along the sampling process by adjusting the $k$ values. The full proof of Theorem 2 and 3 is written in the appendix.

## 4 EXPERIMENTAL RESULTS

**Setup** Experiments are conducted on pretrained Diffusion models on *ImageNet 64x64*, *ImageNet 128x128*, *ImageNet 256x256* and *MSCOCO*. The base Diffusion models utilized for label condition sampling task are ADM Dhariwal & Nichol (2021) and CADM Dhariwal & Nichol (2021) for classifier guidance, DiTPeebles & Xie (2023) for classifier-free guidance (CFG) Ho & Salimans (2022), GLIDENichol et al. (2021) for CLIP text-to-image guidance and Stable Diffusion Rombach

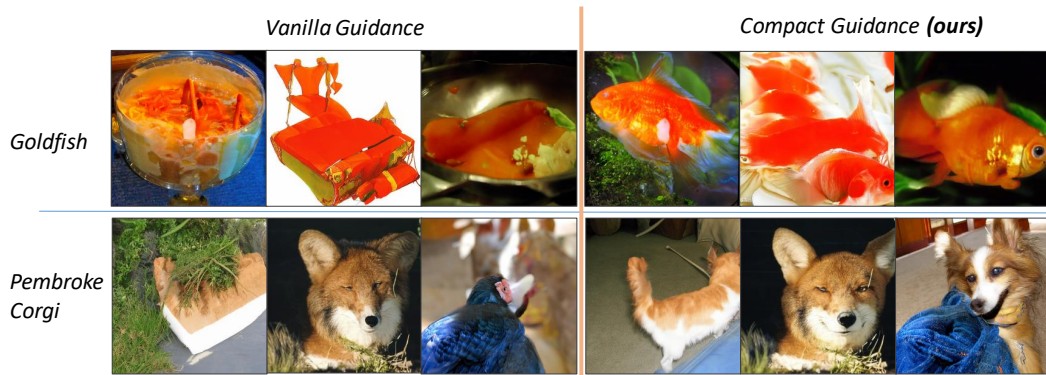

Figure 6: *Qualitative results on ImageNet256x256. Left: Vanilla guidance applied at all timesteps. Right: Compress Guidance applied at 50 of 250 timesteps. Compress Guidance corrects misclassification by the on-sampling classifier, preventing out-of-class image generation and restoring accurate class information. More qualitative results are shown in AppendixH*

et al. (2022) for text-to-image classifier-free guidance. Other baselines we also do comparison is BigGAN Brock et al. (2018), VAQ-VAE-2 Zhao et al. (2020), LOGAN Wu et al. (2019), DCTransformers Nash et al. (2021). FID/sFID, Precision and Recall are utilized to evaluate image quality and diversity measurements. We denote Compress Guidance as "-CompG" and "-G" as vanilla guidance, "-CFG" is the CFG, and "-CompCFG" is our proposed Compress Guidance applying on CFG. Full results with details of the experimental set up are discussed in Appendix B and C.

## 4.1 CLASSIFIER GUIDANCE

For classifier guidance, we distinguish this guidance scheme into two types due to its behaviour discrepancy when applying the guidance. The first type is classifier guidance on the unconditional diffusion model, and the second is classifier guidance on the conditional diffusion model.

**Guidance with unconditional diffusion model** Guidance with unconditional model provides diffusion model both conditional information and image quality improvement Dhariwal & Nichol (2021). Table 3 shows the improvement using CompactGuidance (CG). The results show three main improvements. First, there is an improvement in the quantitative results of FID, sFID, and Recall values, indicating an improvement in generated image qualities and diversity. Second, we further validate the image quality and diversity improvement in Figure 5 and 6. Third, the proposed method offered a significant improvement in running time where we reduced the number of guidance steps by 5 times and reduced the running time by 42% on ImageNet64x64 and 23% on ImageNet256x256.

**Guidance with conditional diffusion model** Unlike the unconditional diffusion model, guidance in the conditional diffusion model does not aim to provide conditional information. Therefore, the effect of guidance is less significant than guidance on the unconditional diffusion model. Table 4 shows the diversity improvement based on Recall values compared to vanilla guidance. Furthermore, CompG reduced the guidance steps by 5 times and reduced the sampling time by 39.79% , 29.63% , and 22% on ImageNet64x64, 128x128 and 256x256, respectively.

## 4.2 CLASSIFIER-FREE GUIDANCE

Classifier-free guidance is a different form of guidance from classifier guidance. Although classifier-free guidance does not use an explicit classifier for guidance, the diffusion model serves as an implicit classifier inside the model as discussed in Appendix E. We hypothesize that classifier-free guidance also suffers from a similar problem with classifier guidance. We apply the Compress Guidance technique on classifier-free guidance (CompCFG) and demonstrate the results in Table 4.

## 4.3 TEXT-TO-IMAGE GUIDANCE

Besides using labels for conditional generation, text-to-image allows users to input text conditions and generate images with similar meanings. This task has recently become one of the most popular tasks in generative models. We apply the CompactGuidance on this task with two types of guidances,

Table 3: *Applying CompG to classifier guidance on unconditional diffusion model. ADM-CompG reduces the number of guidance timesteps by fivefold and increases the sampling process's running time by approximately 42% on ImageNet64x64 and 23% on ImageNet256x256. Notably, on ImageNet256x256, the running time of ADM-CompG is only 5% higher compared to the unguided sampling process. In terms of performance, ADM-CompG significantly outperforms ADM and ADM-G across most metrics.*

| Model | $|G|$ ($\downarrow$) | GPU hours ($\downarrow$) | FID ($\downarrow$) | sFID ($\downarrow$) | Prec ($\uparrow$) | Rec ($\uparrow$) |
|---|---|---|---|---|---|---|
| **ImageNet 64x64** | | | | | | |
| ADM (No guidance) | 0 | 26.33 | 9.95 | 6.58 | 0.60 | 0.65 |
| ADM-G | 250 | 54.86 | 6.40 | 9.67 | **0.73** | 0.54 |
| **ADM-CompG** | **50** | **31.80** | **5.91** | **8.26** | 0.71 | **0.56** |
| **ImageNet 256x256** | | | | | | |
| ADM (No guidance) | 0 | 245.37 | 26.21 | 6.35 | 0.61 | 0.63 |
| ADM-G | 250 | 334.25 | 11.96 | 10.28 | 0.75 | 0.45 |
| **ADM-CompG** | **50** | **258.33** | **11.65** | **8.52** | **0.75** | **0.48** |

Table 4: *Applying CompG to classifier guidance in conditional diffusion models and classifier-free guidance significantly improves performance. CADM-CompG outperforms CADM and slightly surpasses CADM-G, as CADM-G depends on both the classifier and conditional diffusion model. CompG reduces the number of guidance timesteps by fivefold and significantly increases the sampling process's running time across all three ImageNet resolutions. CompG for classifier-free guidance also reduces the number of guidance steps by tenfold and achieves significantly better results.*

| Model | $|G|$ ($\downarrow$) | GPU hours ($\downarrow$) | FID ($\downarrow$) | sFID ($\downarrow$) | Prec ($\uparrow$) | Rec ($\uparrow$) |
|---|---|---|---|---|---|---|
| **ImageNet 64x64** | | | | | | |
| CADM (No guidance) | 0 | 26.64 | 2.07 | 4.29 | 0.73 | 0.63 |
| CADM-G | 250 | 53.52 | 2.47 | 4.88 | **0.80** | 0.57 |
| **CADM-CompG** | **50** | **32.22** | **1.82** | **4.31** | 0.76 | **0.61** |
| CADM-CFG | 250 | 54.97 | 1.89 | 4.45 | **0.77** | 0.60 |
| **CADM-CompCFG** | **25** | **29.29** | **1.84** | **4.38** | **0.77** | **0.61** |
| **ImageNet 128x128** | | | | | | |
| CADM (No guidance) | 0 | 61.60 | 6.14 | 4.96 | 0.69 | 0.65 |
| CADM-G | 250 | 94.06 | 2.95 | 5.45 | **0.81** | 0.54 |
| **CADM-CompG** | **50** | **66.19** | **2.86** | **5.29** | 0.79 | **0.58** |
| **ImageNet 256x256** | | | | | | |
| CADM (No guidance) | 0 | 240.33 | 10.94 | 6.02 | 0.69 | 0.63 |
| CADM-G | 250 | 336.05 | 4.58 | **5.21** | 0.81 | 0.51 |
| **CADM-CompG** | **50** | **259.25** | **4.52** | 5.29 | **0.82** | 0.51 |
| DiT-CFG | 250 | 75.04 | 2.25 | **4.56** | 0.82 | 0.58 |
| **DiT-CompCFG** | **22** | **42.20** | **2.19** | 4.74 | **0.82** | **0.60** |

which are CLIP-based guidance (GLIDE) Nichol et al. (2021) and classifier-free guidance (Stable Diffusion) Rombach et al. (2022). The results are shown in Table 5 and 6 and Figure 1.

Table 5: *Applying CompG on text-to-image GLIDE classifier-based guidance on MSCoco datasets.*

| Model | $|G|$ ($\downarrow$) | GPU hrs ($\downarrow$) | ZFID ($\downarrow$) |
|---|---|---|---|
| **MSCOCO 64x64** | | | |
| GLIDE-G | 250 | 34.04 | 24.78 |
| **GLIDE-CompG** | **25** | **20.93** | **24.5** |
| **MSCOCO 256x256** | | | |
| GLIDE-G | 250 | 66.84 | 34.78 |
| **GLIDE-CompG** | **35** | **37.55** | **33.12** |

Table 6: *Applying CompG on Stable Diffusion classifier-free guidance on MSCoco256x256 dataset. CompG significantly improve both qualitative results, as in Figure 1, and quantitative results, as below.*

| Model | $|G|$ ($\downarrow$) | GPU hrs ($\downarrow$) | FID ($\downarrow$) | IS ($\uparrow$) | CLIP ($\uparrow$) |
|---|---|---|---|---|---|
| **MSCOCO 256x256** | | | | | |
| SD-CFG | 50 | 54 | 16.04 | 32.34 | 28 |
| **SD-ComptCFG** | **8** | **35** | **14.04** | **35.90** | **30** |

## 4.4 ABLATION STUDY

**Solving the model-fitting problem** One of the main contributions of the proposed method is its help in alleviating the model-fitting problem. Due to the closeness be-

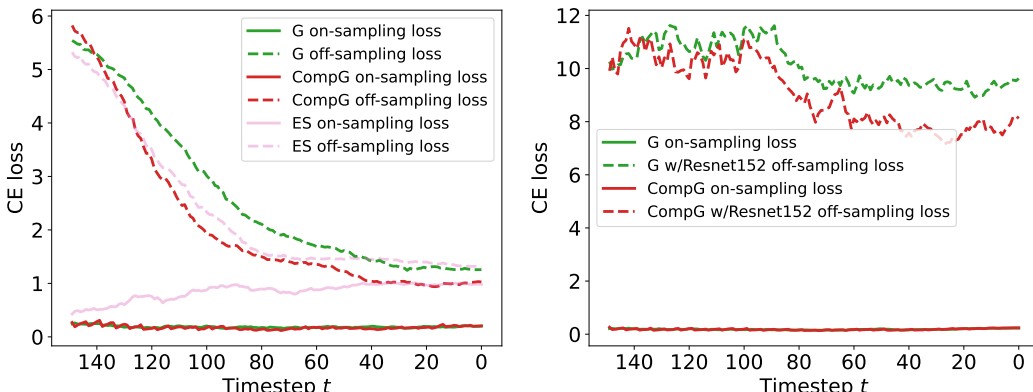

Figure 7: *CompG reduces the gap between off-sampling and on-sampling loss, mitigating the model-fitting issue compared to other schemes. The ES scheme concludes guidance after 50 steps and suffers from forgetting problems where the on-sampling loss increases along with the sampling process.*

tween the model-fitting problem and overfitting problems, we use an Early stopping scheme for comparison. For CompG, we utilize 50 guidance steps. Thus, we also turn off guidance for the ES scheme after 50 guidance calls. Figure 7 for details.

Table 7: *Model-fitting on ImageNet64x64 samples. ES suffers from the forgetting problem and has low performance. CompG achieves higher both on on-sampling and off-sampling acc.*

| Guidance | On-samp. | Off-samp. | Resnet |
|---|---|---|---|
| Vanilla | 90.8 | 62.5 | 34.17 |
| Early Stopping | 63.05 | 55.22 | 33.55 |
| CompG (ours) | **91.2** | **64.2** | **34.93** |

**Distribution guidance timesteps toward the early stage of the process:** According to the Theorem 2, by adjusting $k$, we can distribute the timesteps toward the early stage or the late stage of the sampling process. Table 8 shows the comparison between $k$ values. With $k = 1.0$, guidance steps are distributed uniformly. Larger $k$ results in comparable performance but more fruitful running time and the number of guidance steps.

Table 8: *ImageNet64x64. Experimental results with increasing $k$. According to Theorem 2, increasing $k$ guides distribution towards early timesteps, resulting in comparable performance comparable to full guidance and better than without guidance. This scheme leads to fewer guidance steps and lower running costs.*

| Model | k | $|G|$ ($\downarrow$) | GPU hours ($\downarrow$) | FID ($\downarrow$) | sFID ($\downarrow$) | Prec ($\uparrow$) | Rec ($\uparrow$) |
|---|---|---|---|---|---|---|---|
| CADM (No guidance) | - | 0 | 26.64 | 2.07 | 4.29 | 0.73 | 0.63 |
| CADM-ComptG | 1.0 | 50 | 32.22 | 1.91 | 4.38 | **0.77** | 0.61 |
| CADM-ComptG | 2.0 | 47 | 31.18 | 1.95 | 4.40 | 0.76 | 0.62 |
| CADM-ComptG | 3.0 | 41 | 30.54 | 1.94 | 4.42 | 0.76 | 0.62 |
| CADM-ComptG | 4.0 | 36 | 30.02 | 1.89 | 4.35 | 0.76 | 0.62 |
| CADM-ComptG | 5.0 | 32 | 29.81 | **1.82** | **4.31** | 0.76 | **0.62** |
| CADM-ComptG | 6.0 | 28 | **29.12** | 1.93 | 4.35 | 0.75 | 0.62 |

**Trade-off between computation and image quality** Compact rate is the total number of sampling steps over the number of guidance steps $\frac{T}{|G|}$. The larger the compact rate, the lower the model's guidance, hence the lower running time. Figure 9 shows the effect of using fewer timesteps on IS, FID and Recall as in Figure 9a, 9b and 9c in Appendix.

# 5 CONCLUSION

The paper quantifies the problem of model-fitting, an analogy to the problem of overfitting in training deep neural networks by observing on-sampling loss and off-sampling loss. Compress Guidance is proposed to alleviate the situation and significantly boost the Diffusion Model's performance in qualitative and quantitative results. Furthermore, applying Compress Guidance can reduce the number of guidance steps by at least five times and reduce the running time by around 40%. Broader Impacts and Safeguards will be discussed in the Appendix.

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

## A  BROADER IMPACT AND SAFEGUARD

The work does not have concerns about safeguarding since it does not utilize the training data. The paper only utilizes the pre-trained models from DiT Peebles & Xie (2023), ADMDhariwal & Nichol (2021), GLIDE Nichol et al. (2021) and Stable Diffusion Rombach et al. (2022). The work fastens the sampling process of the diffusion model and contributes to the population of the diffusion model in reality. However, the negative impact might be on the research on a generative model where bad people use that to fake videos or images.

## B  EXPERIMENTAL SETUP

**Off-sampling classifier**: Off-sampling classifier is initialized as the parameters of the on-sampling classifier. We fine-tune the model with 10000 timesteps with the same loss for training the on-sampling classifier. The testing accuracy between the off-sampling classifier and the on-sampling classifier is shown in Table 9

| Evaluation Model | Accuracy |
|---|---|
| *On-sampling classifier* | 64.5% |
| *Off-sampling classifier* | 63.5% |

Table 9: Evaluation of On-sampling classifier and Off-sampling classifier on ground-truth images.

Figure 11 shows all the hyperparameters used for all experiments in the paper. Normally, since we skip a lot of timesteps that do guidance, the process will fall into the case of forgetting. To avoid this situation, we would increase the guidance scale significantly. The value of the guidance scale is often based on the compact rate $\frac{T}{|G|}$. A larger compact rate also indicates a larger guidance scale.

In Table 7 and Figure 7, to achieve a fair comparison, we tune the guidance scale of CompG to achieve a similar Recall value with vanilla guidance. The reason is that the higher the level of diversity, the harder features can be recognized resulting higher loss and lower accuracy. If we don't configure similar diversity between two schemes, the one with higher diversity will always achieve lower accuracy and higher loss value. We want to avoid the case that the model only samples one good image for all.

For all the tables, the models which are in bold are the proposed.

**GPU hours**: All the GPU hours are calculated based on the time for sampling 50000 samples in ImageNet or 30000 samples in MSCoco.

All experiments are run on a cluster with 4 V100 GPUs.

## C  FULL COMPARISION

Table 10 shows full comparison with different famous baselines.

## D  MATHEMATICAL DETAILS

**Proof of Theorem 1**

*Proof.* Given real data $\mathbf{x}_0$, we sample two latent samples at two timestep $t_1 < t_2$. As a result $\mathbf{x}_{t_1} = \sqrt{\bar{\alpha}_{t_1}}\mathbf{x}_0 + \sqrt{1-\bar{\alpha}_{t_1}}\epsilon$ and $\mathbf{x}_{t_2} = \sqrt{\bar{\alpha}_{t_2}}\mathbf{x}_0 + \sqrt{1-\bar{\alpha}_{t_2}}\epsilon$. From $\mathbf{x}_{t_1}$ and $\mathbf{x}_{t_2}$, the prediction of real data has the form of $\tilde{\mathbf{x}}_0^{(t_1)} = \frac{\mathbf{x}_{t_1} - \sqrt{1-\bar{\alpha}_{t_1}}\epsilon_\theta(\mathbf{x}_{t_1},t_1)}{\sqrt{\bar{\alpha}_{t_1}}}$ and $\tilde{\mathbf{x}}_0^{(t_2)} = \frac{\mathbf{x}_{t_2} - \sqrt{1-\bar{\alpha}_{t_2}}\epsilon_\theta(\mathbf{x}_{t_2},t_2)}{\sqrt{\bar{\alpha}_{t_2}}}$ correspondingly. Replace $\mathbf{x}_{t_1}$ and $\mathbf{x}_{t_2}$ with $\mathbf{x}_0$ and $\epsilon$, we have $\tilde{\mathbf{x}}_0^{(t_1)} = \mathbf{x}_0 + \frac{\sqrt{1-\bar{\alpha}_{t_1}}(\epsilon-\epsilon_\theta(\mathbf{x}_{t_1},t_1))}{\sqrt{\bar{\alpha}_{t_1}}}$ and $\tilde{\mathbf{x}}_0^{(t_2)} = \mathbf{x}_0 + \frac{\sqrt{1-\bar{\alpha}_{t_2}}(\epsilon-\epsilon_\theta(\mathbf{x}_{t_2},t_2))}{\sqrt{\bar{\alpha}_{t_2}}}$. Thus $||\tilde{\mathbf{x}}_0^{(t_1)} - \mathbf{x}_0|| = \frac{1-\bar{\alpha}_{t_1}||\epsilon-\epsilon_\theta(\mathbf{x}_{t_1},t_1)||}{\bar{\alpha}_{t_1}}$ and

Table 10: *We show full results of the model compared to other models not related to guidance.*

| Model | $|G|$ ($\downarrow$) | GPU hours ($\downarrow$) | FID ($\downarrow$) | sFID ($\downarrow$) | Prec ($\uparrow$) | Rec ($\uparrow$) |
|---|---|---|---|---|---|---|
| **ImageNet 64x64** | | | | | | |
| BigGAN | - | - | 4.06 | 3.96 | 0.79 | 0.48 |
| IDDPM | 0 | 28.32 | 2.90 | 3.78 | 0.73 | 0.62 |
| CADM (No guidance) | 0 | 26.64 | 2.07 | 4.29 | 0.73 | 0.63 |
| CADM-G | 250 | 53.52 | 2.47 | 4.88 | **0.80** | 0.57 |
| **CADM-CompG** | **50** | **32.22** | 1.91 | **4.57** | 0.77 | **0.61** |
| CADM-CFG | 250 | 54.97 | 1.89 | 4.45 | **0.77** | 0.60 |
| **CADM-CompCFG** | **25** | **29.29** | 1.84 | **4.38** | 0.77 | 0.61 |
| **ImageNet 128x128** | | | | | | |
| BigGAN | - | - | 6.02 | 7.18 | 0.86 | 0.35 |
| LOGAN | - | - | 3.36 | - | - | - |
| CADM (No guidance) | 0 | 61.60 | 6.14 | 4.96 | 0.69 | 0.65 |
| CADM-G | 250 | 94.06 | 2.95 | 5.45 | **0.81** | 0.54 |
| **CADM-CompG** | **50** | **66.19** | 2.86 | 5.29 | 0.79 | **0.58** |
| **ImageNet 256x256** | | | | | | |
| BigGAN | - | - | 7.03 | 7.29 | 0.87 | 0.27 |
| DCTrans | - | - | 36.51 | 8.24 | 0.36 | 0.67 |
| VQ-VAE-2 | - | - | 31.11 | 17.38 | 0.36 | 0.57 |
| IDDPM | - | - | 12.26 | 5.42 | 0.70 | 0.62 |
| CADM (No guidance) | 0 | 240.33 | 10.94 | 6.02 | 0.69 | 0.63 |
| CADM-G | 250 | 336.05 | 4.58 | **5.21** | 0.81 | 0.51 |
| **CADM-CompG** | **50** | **259.25** | 4.52 | 5.29 | **0.82** | 0.51 |
| DiT-CFG | 250 | 75.04 | 2.25 | **4.56** | 0.82 | 0.58 |
| **DiT-CompCFG** | **22** | **42.20** | **2.19** | 4.74 | **0.82** | **0.60** |

$||\tilde{\mathbf{x}}_0^{(t_2)} - \mathbf{x}_0|| = \frac{1-\bar{\alpha}_{t_2}||\epsilon-\epsilon_\theta(\mathbf{x}_{t_2},t_2)||}{\bar{\alpha}_{t_2}}$. Since $\epsilon_\theta(\mathbf{x}_{t_1},t_1) \sim \epsilon_\theta(\mathbf{x}_{t_2},t_2) \sim \epsilon$, $||\epsilon - \epsilon_\theta(\mathbf{x}_{t_1},t_1)|| \approx$ $||\epsilon - \epsilon_\theta(\mathbf{x}_{t_2},t_2)|| \approx \Delta$. This results in $||\tilde{\mathbf{x}}_0^{(t_1)} - \mathbf{x}_0|| = \frac{1-\bar{\alpha}_{t_1}}{\bar{\alpha}_{t_1}}\Delta$ and $||\tilde{\mathbf{x}}_0^{(t_2)} - \mathbf{x}_0|| = \frac{1-\bar{\alpha}_{t_2}}{\bar{\alpha}_{t_2}}\Delta$. $||\tilde{\mathbf{x}}_0^{(t_1)} - \mathbf{x}_0|| < ||\tilde{\mathbf{x}}_0^{(t_1)} - \mathbf{x}_0||$ since $\frac{1-\bar{\alpha}_{t_2}}{\bar{\alpha}_{t_2}} > \frac{1-\bar{\alpha}_{t_1}}{\bar{\alpha}_{t_1}} \geq 0, \forall t_2 > t_1$. As a result, the sampling of $\mathbf{x}_{t-1} \sim q(\mathbf{x}_{t-1}|\mathbf{x}_t,\tilde{\mathbf{x}}_0)$ from timesteps $T$ to 0 would result in the minimization of $||\tilde{\mathbf{x}}_0^{(t)} - \mathbf{x}_0||$. Since $q(\tilde{\mathbf{x}}_0)$ has the form of Gaussian, we can have the minimization of $||\tilde{\mathbf{x}}_0^{(t)} - \mathbf{x}_0||$ would result in the minimization of $||q(\tilde{\mathbf{x}}_0) - q(\mathbf{x}_0)|| = ||\frac{q(\tilde{\mathbf{x}}_0)q(\mathbf{x}_t|\tilde{\mathbf{x}}_0)}{q(\mathbf{x}_t)} - q(\mathbf{x}_0)||$ since $\tilde{\mathbf{x}}_0 \sim p_\theta(\tilde{\mathbf{x}}_0|\mathbf{x}_t)$ with a deterministic forward of $\mathbf{x}_t$ to $\epsilon_\theta$, we have $q(\tilde{\mathbf{x}}_0) \approx \frac{q(\tilde{\mathbf{x}}_0)q(\mathbf{x}_t|\tilde{\mathbf{x}}_0)}{q(\mathbf{x}_t)} = p_\theta(\tilde{\mathbf{x}}_0|\mathbf{x}_t)$.

Assume we have two density function $p(\mathbf{x})$ and $q(\mathbf{x})$. The KL divergence between these two has the form:

$$\int_0^1 p(\mathbf{x}) \log \frac{p(\mathbf{x})}{q(\mathbf{x})} = \int_0^1 p(\mathbf{x}) \log(p(\mathbf{x})) - p(\mathbf{x})\log(q(\mathbf{x}))d\mathbf{x} \tag{16}$$

$$= \int_0^1 p(\mathbf{x})\log(p(\mathbf{x}))d\mathbf{x} - \int_0^1 p(\mathbf{x})\log(p(\mathbf{x})) + p(\mathbf{x})\log((\frac{p(\mathbf{x})}{q(\mathbf{x})}-1)+1)d\mathbf{x} \tag{17}$$

$$= \int_0^1 -p(\mathbf{x})\log((\frac{q(\mathbf{x})}{p(\mathbf{x})}-1)+1)d\mathbf{x} \tag{18}$$

$$= \int_0^1 -(q(\mathbf{x})-p(\mathbf{x})) + (q(\mathbf{x})-p(\mathbf{x}))^2(\frac{1}{p(\mathbf{x})} - \frac{1}{q(\mathbf{x})})d\mathbf{x} \tag{19}$$

$$\leq \int_0^1 (q(\mathbf{x})-p(\mathbf{x}))^2(\frac{1}{p(\mathbf{x})} - \frac{1}{q(\mathbf{x})})d\mathbf{x} \tag{20}$$

$$\leq \int_0^1 (q(\mathbf{x})-p(\mathbf{x}))^2(\frac{1}{a} - \frac{1}{b})d\mathbf{x} = \frac{b-a}{ab}||p-q|| \tag{21}$$

Thus $D_{KL}(p(\mathbf{x})||q(\mathbf{x})) \leq \frac{b-a}{ab}||p-q||$

Base on this bound we would have the minimization of $||p_\theta(\tilde{\mathbf{x}}_0|\mathbf{x}_t) - q(\mathbf{x}_0)||$ is equivalent to the minimization of $D_{KL}(q(\mathbf{x}_0)||p_\theta(\tilde{\mathbf{x}}_0|\mathbf{x}_t))$. □

**Proof of Theorem 2**

*Proof.* Let $k_1 < k_2$ and $k_1, k_2 \in [1; +\infty]$, with $\frac{T}{|G|^k} i^k = T(\frac{i}{|G|})^k$ and $\frac{i}{|G|} < 1$, we have:

$$(\frac{i}{|G|})^{k_1} \geq (\frac{i}{|G|})^{k_2} \tag{22}$$

$$\Leftrightarrow T(\frac{i}{|G|})^{k_1} \geq T(\frac{i}{|G|})^{k_2} \tag{23}$$

$$\Leftrightarrow \lfloor T(\frac{i}{|G|})^{k_1} \rfloor \geq \lfloor T(\frac{i}{|G|})^{k_2} \rfloor \tag{24}$$

$$\Leftrightarrow T - \lfloor T(\frac{i}{|G|})^{k_1} \rfloor \leq T - \lfloor T(\frac{i}{|G|})^{k_2} \rfloor \tag{25}$$

As a result, $G_i^{(k_1)} \leq G_i^{(k_2)} \forall k_1, k_2 \geq 1$ and $k_1 < k_2$. With $k_2 \to +\infty$, $G_i^{(k_2)}$ is bounded by T. This means that larger $k$ values would result in the distribution of the timesteps toward the early stage of the sampling process. □

**Proof of Theorem 3**

*Proof.* Similar to previous proof we have $G_i^{(k_1)} \leq G_i^{(k_2)} \forall k_1, k_2 \geq 1$ and $k_1 < k_2$. This also mean that $G_i^{(k_1)} > G_i^{(1)}$, $\forall 0 \leq k_1 < 1$ and if $k_1 \to 0$ then $G_i^{(k_1)} \to 0$, hence all the $g_i \in G^{(k_1)i}$ is bounded by 0. As a result, by adjusting $k$ toward 0, we would have the distribution of guidance steps toward the later stage of the sampling process □

## E CompG and classifier-free guidance

We start from the noise sampling equation of the classifier-free guidance as:

$$\tilde{\epsilon} = (1 + w)\epsilon_\theta(\mathbf{x}_t, c, t) - w\epsilon_\theta(\mathbf{x}_t, t) \tag{26}$$

$$= \epsilon_\theta(\mathbf{x}_t, c, t) + w(\epsilon_\theta(\mathbf{x}_t, c, t) - \epsilon_\theta(\mathbf{x}_t, t)) \tag{27}$$

$$= \epsilon_\theta(\mathbf{x}_t, c, t) + wC \tag{28}$$

We can clearly see that $C$ stands for classification information as mentioned in Dinh et al. (2023). Replace the $\tilde{\epsilon}$ to Eq.10, we have:

$$\mathbf{x}_{t-1} = \mathbf{x}_t - (\underbrace{\frac{\sqrt{\alpha_t} - 1}{\sqrt{\alpha_t}}\mathbf{x}_t + \frac{1 - \alpha_t}{\sqrt{1 - \bar{\alpha}_t}\sqrt{\alpha_t}}\epsilon_\theta(\mathbf{x}_t, c, t) - \sigma_t \mathbf{z}}_{\text{Original denoising framwork}}) - \underbrace{\frac{\alpha_t - 1}{\sqrt{1 - \bar{\alpha}_t}}wC}_{\text{classification information}} \tag{29}$$

From this derivation, we can further apply the technique from CompG to the classification term in classifier-free guidance.

## F RELATED WORK

Diffusion Generative Models (DGMs) Ho et al. (2020); Song et al. (2020b); Vahdat et al. (2021); Song & Ermon (2020) have recently become one of the most popular generative models in many tasks such as image editingKawar et al. (2023); Huang et al. (2024), text-to-image sampling Rombach et al. (2022) or image generation. Guidance is often utilized to improve the performance of DGMs Dhariwal & Nichol (2021); Ho & Salimans (2022); Bansal et al. (2023); Liu et al. (2023); Epstein et al. (2023). Besides improving the performance, the guidance also offers a trade-off between image quality and diversity [], which helps users tune their sampling process up to their expectations. Although guidance is beneficial in many forms, it faces extremely serious drawbacks of running time. For classifier guidance, the running time is around 80% higher compared to the

original diffusion model sampling time due to the evaluation of gradients at every sampling step. In contrast, classifier-free guidance requires the process to forward to the expensive diffusion model twice at every timestep.

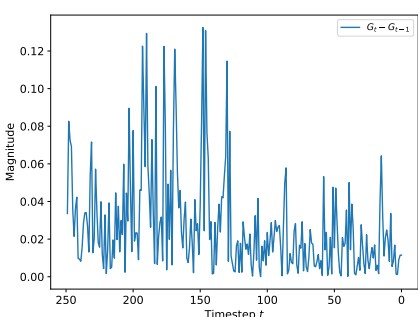

Previous works on improving the running time of DGMs involve the reduction of sampling steps Song et al. (2020a); Zhang & Chen (2022) and latent-based diffusion models Rombach et al. (2022); Peebles & Xie (2023). Recently, the research community has focused on distilling from a large number of timesteps to a smaller number of timesteps Salimans & Ho (2022); Sauer et al. (2023); Li et al. (2024) or reducing the architectures of diffusion models Li et al. (2024). However, most of these works mainly solve the problem of the expensive sampling of diffusion models. As far as we notice, none of the works have dealt with the exorbitant cost resulting from guidance.

Figure 8: *Gradient magnitude difference measured at two consecutive steps*

## G  GRADIENT MAGNITUDE DIFFERENCE BETWEEN TWO CONSECUTIVE SAMPLING STEPS

In this section, we observe that the classification gradient will likely vary significantly in the early stage of the sampling process. We sample 32 images of ImageNet64 using ADM-G (Dhariwal & Nichol (2021)) with guidance classifier is the noise-aware trained classifier from ADM-G. The observation is shown in Fig 8.

## H  ADDITIONAL QUALITATIVE RESULTS

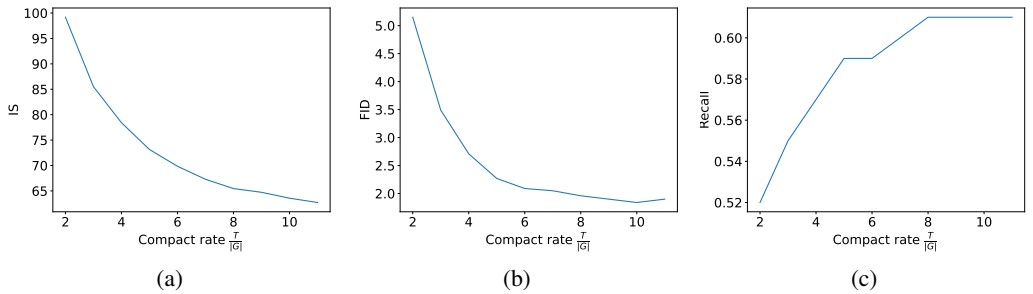

Figure 9: Trade-off: Running time versus performance. We measure the compact rate as $\frac{T}{|G|}$. In (a), IS decreases with increasing compact rate, while FID and Recall improve. However, when the rate exceeds 10, FID begins to rise. This suggests that increased diversity from more features initially enhances Recall and FID, but excessive diversity degrades image quality.

Table 11: All hyper-parameters required for reproducing the results.

| MODEL | DATASET | $k$ | $s$ | $|G|$ | TIME-STEPS |
|---|---|---|---|---|---|
| TABLE 2 | | | | | |
| ADM | IMAGENET 64x64 | 1.0 | 0.0 | 0 | 250 |
| ADM-G | IMAGENET 64x64 | 1.0 | 4.0 | 250 | 250 |
| ADM-COMPG | IMAGENET 64x64 | 1.0 | 4.0 | 50 | 250 |
| ADM | IMAGENET 256x256 | 1.0 | 0.0 | 0 | 250 |
| ADM-G | IMAGENET 256x256 | 1.0 | 4.0 | 250 | 250 |
| ADM-COMPG | IMAGENET 256x256 | 1.0 | 4.0 | 50 | 250 |
| TABLE 3 | | | | | |
| CADM | IMAGENET 64x64 | 1.0 | 0.0 | 0 | 250 |
| CADM-G | IMAGENET 64x64 | 1.0 | 0.5 | 250 | 250 |
| CADM-COMPG | IMAGENET 64x64 | 1.0 | 2.0 | 50 | 250 |
| CADM-CFG | IMAGENET 64x64 | 1.0 | 0.1 | 250 | 250 |
| CADM-COMPCFG | IMAGENET 64x64 | 1.0 | 0.1 | 25 | 250 |
| CADM | IMAGENET 128x128 | 0.9 | 0.0 | 0 | 250 |
| CADM-G | IMAGENET 128x128 | 1.0 | 0.5 | 250 | 250 |
| CADM-CFG | IMAGENET 128x128 | 1.0 | 0.5 | 250 | 250 |
| CADM | IMAGENET 256x256 | 1.0 | 0.0 | 0 | 250 |
| CADM-G | IMAGENET 256x256 | 1.0 | 0.5 | 250 | 250 |
| CADM-COMPG | IMAGENET 256x256 | 1.0 | 0.5 | 50 | 250 |
| DIT-CFG | IMAGENET 256x256 | 1.0 | 1.5 | 250 | 250 |
| DIT-COMPCFG | IMAGENET 256x256 | 1.0 | 1.5 | 22 | 250 |
| TABLE 4 | | | | | |
| GLIDE-G | MSCOCO 64x64 | 1.0 | 0.0 | 250 | 250 |
| GLIDE-COMPG | MSCOCO 64x64 | 1.0 | 8.0 | 25 | 250 |
| GLIDE-G | MSCOCO 256x256 | 1.0 | 0.0 | 250 | 250 |
| GLIDE-COMPG | MSCOCO 256x256 | 1.0 | 5.5 | 35 | 250 |
| TABLE 4 | | | | | |
| SDIFF-CFG | MSCOCO 64x64 | 1.0 | 2.0 | 250 | 250 |
| SDIFF-COMPCFG | MSCOCO 64x64 | 1.0 | 2.0 | 8 | 250 |

Quiet forest path surrounded by tall trees.

Beach at sunset with waves gently crashing.

StableDiffusion      **(ours)**

Figure 10: *Stable Diffusion with classifier-free guidance. The left figure is the vanilla classifier-free guidance with application on all 50 timesteps. Our proposed Compress Guidance method is the right figure, where we only apply guidance on 10 over 50 steps. The output shows our methods' superiority over classifier-free guidance regarding image quality, quantitative performance and efficiency.*

Serene mountain landscape with a clear sky

A white plate with breakfast foods on it

StableDiffusion

**(ours)**

Figure 11: *Stable Diffusion with classifier-free guidance. The left figure is the vanilla classifier-free guidance with application on all 50 timesteps. Our proposed Compress Guidance method is the right figure, where we only apply guidance on 10 over 50 steps. The output shows our methods' superiority over classifier-free guidance regarding image quality, quantitative performance and efficiency.*

Flowers are arranged in a vase sitting on a table.

A plate with food on it, a fork and some kind of drink

StableDiffusion      **(ours)**

Figure 12: *Stable Diffusion with classifier-free guidance. The left figure is the vanilla classifier-free guidance with application on all 50 timesteps. Our proposed Compress Guidance method is the right figure, where we only apply guidance on 10 over 50 steps. The output shows our methods' superiority over classifier-free guidance regarding image quality, quantitative performance and efficiency.*

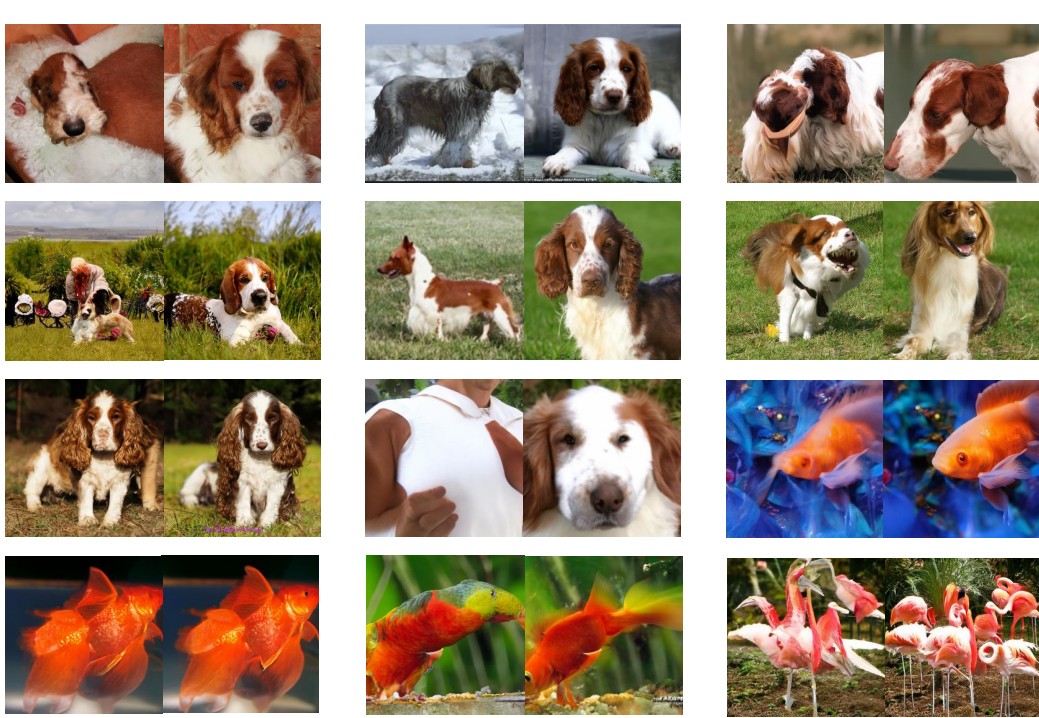

Figure 13: *Qualitiative comparison between ADM-G and ADM-CompG.The image generated by ADM-G and ADM-CompG are put side by side. On the left side is ADM-G and on the right side is ADM-CompG.*

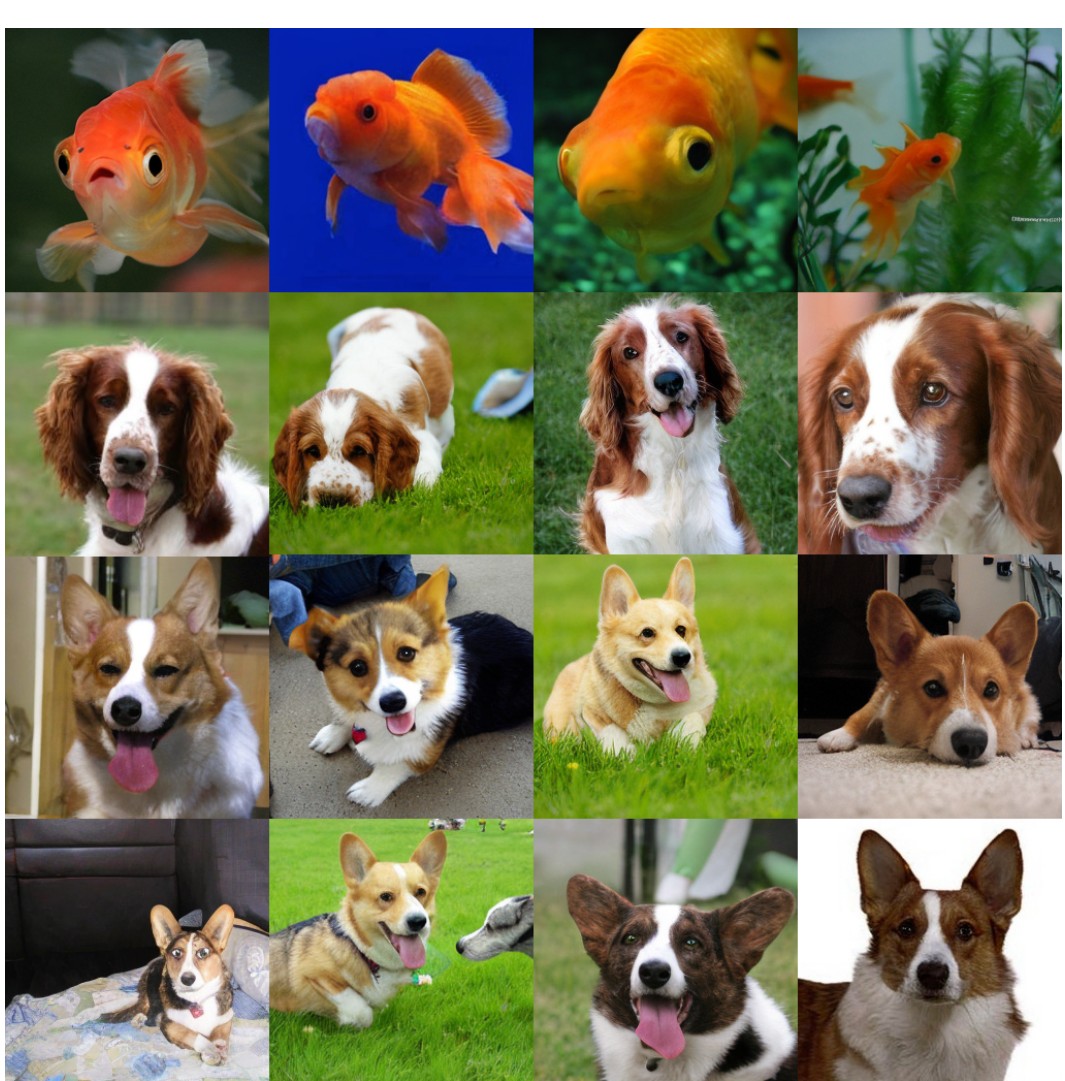

Figure 14: *Images generated by DiT-CompCFG. From top to bottom classes goldfish, Welsh springer spaniel, Pembroke Welsh corgi, Cardigan Welsh corgi.*

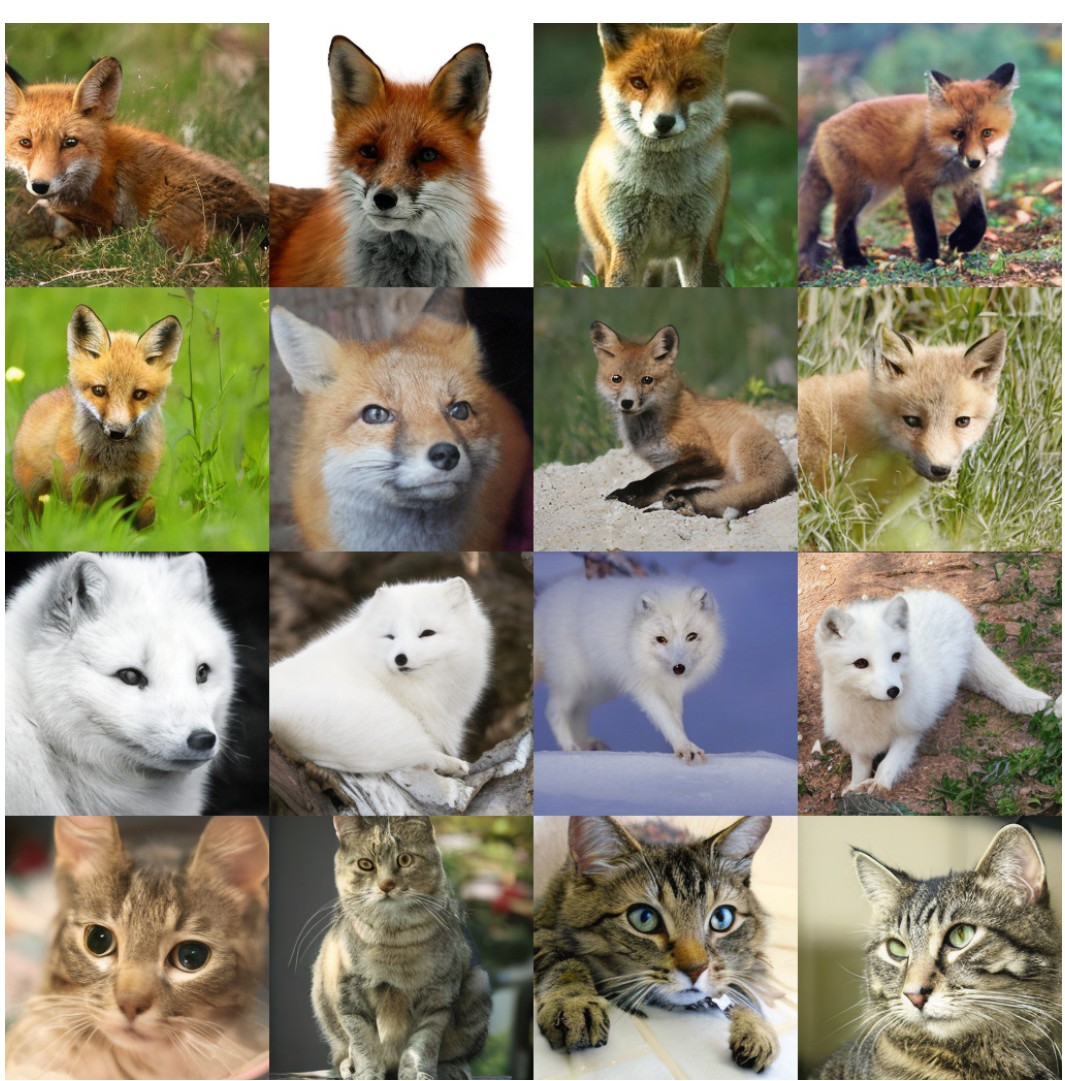

Figure 15: *Images generated by DiT-CompCFG. From top to bottom classes redfox, kitfox, Arctic fox, tabby cat.*

