# OpenReview forum: "Compress Guidance in Conditional Diffusion Sampling"
_ICLR.cc/2025/Conference — ICLR 2025 Conference Withdrawn Submission_

### Official Review · Reviewer_f7Bg · 2024-10-30

**Soundness:** 1
**Presentation:** 2
**Contribution:** 1
**Rating:** 3
**Confidence:** 5

**Summary:**

The paper analyzes the overfitting problem of classifier gradient in guided sampling of diffusion models, claiming that guidance at every denoise step might harm the conditional fidelity. In order to avoid too much gradient, compared to existing Early Stopping or Uniform Skipping techniques, the authors propose a novel trick, named as Compress Guidance, providing sufficient guidance resembling momentum method.

**Strengths:**

- The paper is well structured, and the motivation is clear.

- The comparison by employing Early Stopping and Uniform Skipping is quite intuitive and easy to follow.

- The novelly proposed Compress Guidance encourages further works delving into guided sampling theory.

**Weaknesses:**

- There are plenty of theoretical flaws in the paper, and some conclusions are not that obvious to draw but with no clarification.
  1. Proof of Thm. 1 is wrong. Forward process of diffusion model is technically a Markovian process, therefore, one cannot assume that two $\mathbf{x}_t$ at different timesteps $t_1$ and $t_2$ are diffused with the same noise $\epsilon$. Besides, noise prediction is not independent with $\epsilon$, so one cannot assume that $|\epsilon - \epsilon(\mathbf{x}_t,t)|$ has consistent approximate $\Delta$ at different $t$.
  2. Conclusion in Thm. 1 cannot be generalized to any data distribution. In L144, $\epsilon(\mathbf{x}_t,t)\sim\epsilon$ is wrong in general case. Noticeably, $\epsilon(\mathbf{x}_t,t)=\mathbb{E}[\epsilon|\mathbf{x}_t]$, and no further evidence on the distribution of noise prediction. Besides, one can easily calculate the score function (and hence the ground-truth noise prediction $\epsilon(\mathbf{x}_t,t)$) under Gaussian Mixture setting, e.g., $\mathbf{x}_0\sim\mathcal{N}(1,1)+\mathcal{N}(10,1)$.
  3. I cannot understand Eq. (10), why the term is proportional to gradient of KL divergence? There is no detailed explanation for L193, why is the full form added with a coefficient $q(y)$ and why is it equivalent to another gradient?

- The analysis in Sec. 3.1 is less convincing due to logical error. If there is model-fitting problem, as the paper claims, why the conclusion from off-sampling loss is credible? What if on-sampling classifier is somewhat more ground-truth but the off-sampling loss is wrong caused by the model-fitting problem on the off-sampling classifier? Or in other words, if classifiers are not credible, all conclusions in Sec. 3.1 are not credible since they are drawn by analysis using classifiers.

- The whole pipeline makes no sense to me. As stated in Sec. 3.1, guidance using gradient from classifier may be harmful since model-fitting problem on classifiers. Then why using gradient from previous steps will be less harmful? The denoising process still employs gradient guidance at every step, and I cannot tell the superiority of involving previous step or a summation. Why the classification result at $t=1000$ still works for $t=950$?

- The experimental results are also not that convincing.
  1. First, if the on- and off-sampling analysis are correct, then ES most reduces the gap between on- and off-sampling loss, indicating it outperforms the proposed method. Besides, why use 150 NFEs rather than 250 like before? Why ES on-sampling loss does not reduce first and then increase as in Fig. 4? The authors may need further clarification.
  2. Second, visualization in Fig. 6 fails to demonstrate the outperformance, where the novel method generates samples with obvious artifacts.
  3. There are no comparison on FID between CompG, ES and UG. Also will it still be the case when using NFE = 50, 25 or even less?

- The writing is poor and hard to read with too many typos: L90 missing $t$ in subscript, L96 no right parenthesis, Eqs. (5,6) should be $\sigma_t^2\mathbf{I}$, L120 no $t$ in $\epsilon_\theta$, L159 missing parentheses.

**Questions:**

- As stated in Weaknesses, could the authors make it more detailed in Sec. 3 the theory part, especially gradient of KL divergence?

---

### Official Review · Reviewer_upNj · 2024-11-03

**Soundness:** 3
**Presentation:** 2
**Contribution:** 3
**Rating:** 6
**Confidence:** 4

**Summary:**

This paper analyzes the model-fitting problem and finds that the relatively less guidance for diffusion is more important for model performance.

This paper proposes a simple yet quite effective method to address model-fitness.

This paper also provides detailed experimental results to validate their hypothesis.

Apart from the performance, the method shows faster convergence speed and save training cost.

**Strengths:**

1. This work provides clear theoretical and experimental explanations for model-fitting. The explanation is convincing.

2. The method is simple but effective, it is not hard to implement technically.

3. This paper includes sufficient experiments including U-Net and Transformer-based Diffusion models.

**Weaknesses:**

1. As the key contribution and observation of this paper, the experiments of model-fitting shown in Figure 2 are not well explained (such as the details of the model you use, and the dataset setting). Moreover, the main concern is whether the conclusion from Figure 2 is still valid on different model architectures and different datasets.

2. The main assumption of this method is that the gradient of guidance should be concentrated in the early stages. Ignoring the latter stage guidance means less detailed information relative to the class will be present in the final images. But in the images with more detailed information, this will lose more fine-grained elements in the images. What's your comment on this?

3. Based on the above discussion, I think a trade-off curve of the skipped steps(or GPU hours to model converge) and the final performance should be included in the paper.

**Questions:**

See above.

---

### Official Review · Reviewer_kim7 · 2024-11-04

**Soundness:** 3
**Presentation:** 3
**Contribution:** 2
**Rating:** 3
**Confidence:** 2

**Summary:**

This paper reveals that enforcing guidance throughout the sampling process can be counterproductive. It identifies a model-fitting issue where samples are tuned to match classifier parameters instead of generalizing the expected condition. It shows that reducing or excluding guidance at numerous timesteps can mitigate this problem. By distributing a small amount of guidance over many sampling timesteps, the authors observe significant improvements in image quality and diversity. Their proposed method, Compress Guidance, reduces required guidance timesteps by nearly 40% while surpassing baseline models in image quality.

**Strengths:**

Pros:

- The paper addresses a significant issue: guidance in diffusion models.
- The experiments are well-structured and organized.

**Weaknesses:**

Cons:

- The motivation for compressing guidance is unclear. The paper fails to adequately demonstrate through experiments the weaknesses of uncompressed guidance, such as model-fitting issues and poor image quality.
- Distributing guidance across different timesteps presents a vast search space, which is a significant challenge.
- The method described in section 3.3 doesn't seem as simple as claimed in the paper's contributions.
- The table format is uncomfortable to read and appears inconsistent with the ICLR template and other papers.

Minor issue: There's a missing citation on line 808.

**Questions:**

as above

---

### Official Review · Reviewer_dhbC · 2024-11-04

**Soundness:** 2
**Presentation:** 1
**Contribution:** 2
**Rating:** 3
**Confidence:** 4

**Summary:**

The authors propose the model-fitting problem in conditional diffusion models and propose a solution for model-fitting using compress guidance.

**Strengths:**

How they set up the problem of model fitting and on-sampling/off-sampling loss can be novel, but with current presentation, it is hard to understand.

**Weaknesses:**

The paper seems to be written in haste. There are too many typos and grammatical errors which hurts reading. A lot of references of table and figure are mis-referenced, which adds another barrier of difficulty. A lot of quotation marks are wrong.

The main idea they propose here is model fitting, but reading their definition, I can't quite get what they exactly mean by model fitting. It is too loosely defined. Also, in off-sampling loss, I don't know what phi' is and how it's obtained. With this missing insight, I don't know how to interpret their Table 1, Figure 2, and evidence for model fitting.

I also find they are lacking in literature review. The role of CFG and its improved version of it have been studied extensively recently, e.g., [1-4]. Especially, [3] and [4] discuss CFG scheduling, which is very related to what they're doing in Compress Guidance.


[1] Chung, Hyungjin, et al. "CFG++: Manifold-constrained Classifier Free Guidance for Diffusion Models." arXiv preprint arXiv:2406.08070 (2024).
[2] Sadat, Seyedmorteza, et al. "No Training, No Problem: Rethinking Classifier-Free Guidance for Diffusion Models." arXiv preprint arXiv:2407.02687 (2024).
[3] Wang, Xi, et al. "Analysis of Classifier-Free Guidance Weight Schedulers." arXiv preprint arXiv:2404.13040 (2024).
[4] Yoon, Youngseok, et al. "Model Collapse in the Self-Consuming Chain of Diffusion Finetuning: A Novel Perspective from Quantitative Trait Modeling." arXiv preprint arXiv:2407.17493 (2024).

**Questions:**

In Evidence  3, all three rows latch onto the orange color as an important feature. It is counterintuitive that with more guidance, the image starts to become more different from the intended object.

On-sampling loss and Off-sampling loss are defined with classifier parameters. How are they relevant to classifier-free guidance?

---

### Official Review · Reviewer_z6tK · 2024-11-04

**Soundness:** 3
**Presentation:** 2
**Contribution:** 2
**Rating:** 6
**Confidence:** 4

**Summary:**

This paper identifies a model fitting problem in classifier guidance used in conventional diffusion models and proposes a solution called compress guidance. The model fitting issue is illustrated by showing that while the on-sampling loss decreases, the off-sampling loss trend differs completely, indirectly demonstrating that sampling is fit to the parameters of the guidance model. Compress guidance addresses this by compressing the duplicated gradients, mitigating the fitting problem.

**Strengths:**

1.	The paper presents the model fitting problem of classifier guidance with experimental evidence.
2.	It proposes various alternative methods to address this problem, demonstrating that the proposed compress guidance is the most effective.
3.	The approach shows numerical improvements across various models and guidance scenarios.

**Weaknesses:**

1.	Lack of analysis regarding the relationship with the ODE sampler: This method inherently requires more sampling steps to function effectively. A performance comparison based on SNR variation via the ODE sampler seems necessary.

**Questions:**

Could you provide the full algorithm?

---

### Note · Authors · 2024-11-15

**Comment:**

Dear reviewers,

We thank the reviewers' efforts in reviewing our paper.

We generally agree that due to the limited space, we present the model-fitting problem too densely, confuses reviewers dhbC and kim7, and actually, a number of details have been left in the Appendix that answer most of the questions from upNj and j7Bg. While I mostly do not agree with the interpretation of reviewer j7Bg about our theoretical part, we agree that the writing of the notation, especially the $\epsilon$, should be re-wrote carefully to avoid confusion.

Your comments help us to improve the manuscripts for the next submission.


Best regards,

**Withdrawal Confirmation:**

I have read and agree with the venue's withdrawal policy on behalf of myself and my co-authors.